# Influence of Executive Function Training on BMI, Food Choice, and Cognition in Children with Obesity: Results from the TOuCH Study

**DOI:** 10.3390/brainsci13020346

**Published:** 2023-02-17

**Authors:** Sandra Luis-Ruiz, Cristina Sánchez-Castañeda, Maite Garolera, Sara Miserachs-González, Marta Ramon-Krauel, Carles Lerin, Consuelo Sanchez, Núria Miró, Sònia Martínez, Maria Angeles Jurado

**Affiliations:** 1Departament de Psicologia Clínica i Psicobiologia, Institut de Neurociències (UBNeuro), Universitat de Barcelona (UB), 08035 Barcelona, Spain; 2Institut de Recerca Sant Joan de Déu, Esplugues de Llobregat, 08950 Barcelona, Spain; 3Neuropsychology Unit, Consorci Sanitari de Terrassa, 08227 Terrassa, Spain; 4Endocrinology Department, Hospital Sant Joan de Déu, 08950 Esplugues de Llobregat, Spain; 5Paediatric Endocrinology Unit, Consorci Sanitari de Terrassa, 08227 Terrassa, Spain; 6Diabetes Education Unit, Consorci Sanitari de Terrassa, 08227 Terrassa, Spain; 7Pharmacy and Nutrition Unit, Consorci Sanitari de Terrassa, 08227 Terrassa, Spain

**Keywords:** childhood obesity, cognitive training, executive function, quality of life

## Abstract

Background: Children with obesity have a higher risk of future health and psychological problems. Executive functions (EFs) play a key role in successful dietetic and exercise planning; therefore, new treatments aimed at improving EFs may optimize outcomes. Objectives: This study evaluates the impact of EF training on body mass index (BMI), food choice, and cognition in children with obesity. We also examine their real-life executive functioning, emotional state, and quality of life. Methods: Randomized controlled double-blind trial. Forty-six children with obesity were randomly allocated into an executive functions training or a control task training group and attended 30–45 min of daily training (5/week over 6 weeks), with both groups receiving counseling on diet and wearing an activity/sleep tracker. Participants were evaluated at baseline and after treatment. Results: BMI decreased over time in the whole sample, although there were no differences between groups at post-training in BMI, food choice, and cognition. Both groups showed significant improvements in attention, speed, cognitive flexibility, and inhibitory control. Additionally, there were some benefits in real-life executive functioning and self-esteem. Over the 6 weeks, participants showed worse food choices in both groups. Conclusions: EFs training showed a lack of significant effects. The executive function enhancement alone did not explain these changes, as there were no significant differences between the experimental groups. It might be that the control task training could also produce some benefits, and multi-component interventions might be useful for weight loss.

## 1. Introduction

Worldwide obesity has nearly tripled over the past three decades. Prevalence in youth has risen dramatically from 4% in 1975 to more than 18% in 2016 and 39 million children under the age of 5 were overweight or obese in 2020. Children with obesity have a higher risk of future obesity, are more likely to develop non-communicable diseases such as type-2 diabetes and cardiovascular diseases, have disability in adulthood, and premature death [1]. It is considered one of the most serious public health challenges of the 21st century. Furthermore, childhood obesity is also related to psychosocial and psychological problems (i.e., anxiety and depression, low self-esteem, bullying and stigma, and poorer quality of life) [2,3,4,5].

Classic behavioral treatments for obesity have shown several limitations such as long-term failure in maintaining weight loss [6], highlighting the need for novel treatment approaches. Thus, a better understanding of vulnerability factors related to weight gain should help to design more effective treatments [7]. While the basic driver of weight gain is obvious—an energy imbalance between calories ingested and expended—causes are multifactorial and complex [8]. Social factors are directly related to changes in prevalence rates [1], but eating and physical behaviors are also explained by individual factors. Recent evidence suggests that cognitive functions are important determinants of people’s responses to food stimuli and eating behavior [9], and play a key role in successful dietetic and exercise planning [10,11]. Self-control models in adults suggest that the capacity to resist an immediate reward in favor of longer-term goals depends on a balance between two neural systems: (a) an executive decision system involved in impulse control, associated with lateral and medial prefrontal regions, and (b) a reward system that computes the value of a stimulus, associated with the orbitofrontal/ventromedial prefrontal cortex and the striatum [9]. An imbalance between these systems may explain deregulated eating behaviors, as the executive decision system may fail to inhibit the response to rewarding stimuli. There is also some evidence supporting self-control models in children and adolescents. Van den Berg et al. [12] have shown that higher levels of impulsivity are associated with sensitivity to reward, and both aspects are related to overeating [12]. Other authors revealed that reward sensitivity is associated positively with unhealthy snacking [13], fast-food consumption [14,15,16], and even higher body mass index (BMI) [17]. Moreover, in individuals with obesity, there is some evidence of a negative relationship between body-weight status and executive function (EF), attention, and even motor skills [11]. Among all the executive domains assessed, inhibitory control is the most consistently reported to be impaired, although there is certain support for reward sensitivity, attention/set-shifting, and working memory impairments [18].

Different authors have tried to improve executive functioning and influence eating behavior in children through different strategies; such as multicomponent behavioral interventions, physical activity programs, episodic future thinking, and cognitive training [19]. It seems that enhancing executive skills can optimize weight loss treatment outcomes. In fact, a recent review that addressed exploring the impact of cognitive training on cognitive function and real-life behavior (i.e., eating behavior) found some near-transfer benefits of cognitive training to cognitive measures, with inconsistent results regarding far-transfer effects to other real-life measures such as eating behavior [20]. Whereas near-transfer occurs when the training and the outcome task are identical or closely related, far-transfer refers to an improvement in different cognitive skills or different outcomes or tasks [21]. Authors conclude that computerized cognitive training may be a potential weight-loss treatment option, but more research is needed to determine the specific characteristics to enhance treatment outcomes. Topics including the type of training (generalized vs. cue-specific interventions) or which type of stimuli works better (i.e., food-related non-food-related stimuli) are still unclear [20].

To date, only two studies have conducted targeted cognitive training in children and adolescents with obesity and their results are unconclusive [22,23]. Verbeken et al. [22] found BMI maintenance at 2 months follow-up after a cognitive training intervention, while a second study by the same authors [23] did not detect any effects on BMI within the same time frame. For this reason, we conducted this project with the main purpose of evaluating the impact of executive function training on BMI, food choice, and cognition in children with obesity. We also focused on the potential effect of the training on their real-life executive functioning, emotional state, and quality of life, as these are problems frequently related not only to childhood obesity [3,4,5,24,25] but also to executive functions [26,27,28,29]. We hypothesize that children with obesity undergoing the cognitive training program will perform better than active controls in cognitive measures, take better food-related decisions and, consequently, improve their emotional state and quality of life measures at the end of the intervention. We also expect them to maintain their BMI.

## 2. Materials and Methods

### 2.1. Study Design

The study was designed as a randomized controlled double-blind trial (NCT03615274). Patients were randomly allocated by the project coordinator (C.S.C.) according to a sequential number into the ‘executive functions training’ or ‘control task training’ groups after written informed consent was given by parents and verbal consent by children when they came in person to the first assessment session. Randomization was stratified by sex, age, and manual dominance. Assessments were performed by an independent psychologist (S.L.R.) and by a pediatrician, both blind to allocation. Patients and families were also blind to the allocation. The study was approved by the University of Barcelona Institutional Review Board (IRB00003099 protocol 122/V/2016), and by the Consorci Sanitari de Terrassa (02-17-503-039) and Hospital Sant Joan de Déu (PIC-02-19) Review Boards. More details of the study design can be found in Sanchez-Castañeda et al. [30].

### 2.2. Participants

Forty-six children with obesity were recruited from two hospitals located in the area of Barcelona: Consorci Sanitari de Terrassa and Hospital Sant Joan de Déu. Inclusion criteria were having obesity (BMI standard deviation >2 for age and sex) [31]; an age range between 9 and 12 years, including a homogeneous sample of preadolescents (to avoid the confounding factors induced by puberty), and having an intelligence quotient (IQ) within the normal range (80–120). Exclusion criteria were having a neurological, psychiatric, or developmental disorder, secondary obesity, and/or being under treatment with psychotropic drugs. 

A total of 36 children completed the training (>75% of sessions completed) and post-training assessment and were therefore included in our per-protocol analysis. Figure 1 shows the flow diagram of the study recruitment with the sample according to CONSORT reporting guidelines [32].

#### Sample Size

Previous literature on the field has shown significant results with similar sample sizes. A detailed review of appropriate sample sizes to evaluate changes in BMI in children with obesity can be found in our previous publication [30].

### 2.3. Intervention

#### 2.3.1. Cognitive Training

Participants underwent 6 weeks of home-based iPad executive function training (5 sessions/week, 30–45 min/session). Motivation for treatment was assessed before they started by a Likert-type rating scale from 1 (not at all) to 7 (very much).

Participants in the experimental group underwent executive function training, which consisted of working memory training by the Cogmed software and executive function training (mainly inhibitory control, decision-making, and cognitive flexibility) by Cognifit software, both with adaptive difficulty. Cogmed is one of the five commercial cognitive-training programs whose effectiveness has been assessed by several publications [21,33]. Cognifit has no independent studies to date including the pediatric population, although there are some on adults [34,35]. For more details of the evidence of the efficacy of these programs, see Sanchez-Castañeda et al. [30].

Participants of the active control group underwent the control task training, which consisted of the same exercises in Cognifit software but minimizing the executive component and without increasing difficulty (5 sessions/week, 30–45 min/session). We did not use control task training for Cogmed, since the platform no longer provides non-adaptive training.

The research team was in contact with parents through mobile phones once or twice a week, to check that there were no problems with training compliance and that they sent the required and reliable information, or to provide assistance if they experienced technical issues or any other reason that made the daily training difficult (i.e., illness).

#### 2.3.2. Psychoeducation and Food Register

Participants received dietary and healthy lifestyle counseling using animated daily presentations displayed on Prezi (www.prezi.com, accessed on 30 January 2018), which included a variety of content such as healthy food recommendations, healthy-fun recipes (i.e., visually attractive: representing animals, cartoons, etc.), enhancing physical activity, and strategies to manage emotion and behavior. 

In addition, participants were required to provide daily pictures of their food intake, sending photos of the meals through the iPad. We considered valid weekly registers the ones with 4 or more days with complete intake pictures (≥4 meals), from which one should be a weekend day.

#### 2.3.3. Activity and Sleep Pattern Feedback

Participants were provided with a Fitbit Flex 2 (https://www.fitbit.com/flex2) to monitor physical activity and sleep patterns during the weeks of training. They could check their tracking on the iPads and Fitbit sent them badges and positive-feedback messages when they reached a milestone.

### 2.4. Outcomes

Outcomes were collected at baseline (T0), at post-training (T1; 6 weeks after baseline), and during the training period. The assessment consisted of:

#### 2.4.1. Pediatric Assessment

Personal and familial medical history, anthropometric measures (height, weight, waist circumference), and food-choice habits (original Kidmed questionnaire) [36] were assessed at T0 and T1 and considered as far-transfer primary measures. 

#### 2.4.2. Neuropsychological Assessment

Verbal and visual IQ was estimated by the vocabulary and the matrix reasoning subtests of the Wechsler Intelligence Scale for Children—Fifth edition (WISC-V, Spanish version) [37] to control for potential differences in IQ between groups at baseline. 

Socioeconomic data from family and school information were also obtained to characterize the sample. Socioeconomic data were computed by considering familiar annual incomes and parents’ educational level and profession.

According to our main purpose, the following cognitive domains were evaluated at T0 and T1:

Attention and speed: measured by the forward digit span of WISC-V, the forward spatial span of Wechsler non-verbal (WNV) [38], the part I of Children’s Color Trail Test (CCTT) [39], the reading and counting subtests of Five Digits Test (FDT)002C [40], and the Conners’ Continuous Performance Test 3 (CPT3) [41].

Executive functions: We evaluated the three core subdomains proposed by Diamond [28]: (a) cognitive flexibility was evaluated by the shifting subtest of FDT and part II of CCTT; (b) working memory was assessed by the backward digit span of WISC-V, the backward spatial span of WNV, and custom N-back task [42]; and (c) inhibitory control was assessed by the choosing subtest of FDT and a go–no go task [43]. Planning was also measured, using the Tower of London test [44].

All cognitive scores were z-scaled and then added into a global composite considered as a near-transfer primary measure. Attention and executive function subdomains were further analyzed by a post hoc analysis with Bonferroni correction for multiple comparisons. For more details on the neuropsychological assessment, see Sanchez-Castañeda et al. [30].

#### 2.4.3. Rating Scales (Quality of Life, Emotional, and Behavioral Measures)

We assessed executive function behaviors by the Behavior Rating Inventory of Executive Function 2 (BRIEF-2) [45]. Quality of life was evaluated by the Pediatric Quality of Life Inventory (PedsQL v4.0) [46]. Children’s self-esteem and perceived social support were assessed by the Self-Perception and Social Support Profile for Children (SPPC) test [47]. Internalizing and externalizing symptoms were assessed by the Child Behavior Checklist (CBCL) [48]. All rating scale measures were further analyzed by a post hoc analysis and considered far-transfer secondary outcomes. For more details, see Sanchez-Castañeda et al. [30].

#### 2.4.4. Food Choice

Pictures of the daily intake were collected in tables to assess the type of food per day and week. Weekly data from intake pictures were rated by two assessors (S.L.R. and S.M.G.) using an in-house modified version of the Kidmed questionnaire [36]. The original questionnaire does not allow the quantification of meat, eggs, potato, processed food, and extra intake. Therefore, we extended the questionnaire, according to our nutritionist counseling (S.M.), and included 7 items that collected and quantified these categories. In the new custom scale, called from now on Kidmed-modified, the total values range from 0 to 15, with higher scores meaning better food choices. Inter-rater assessment consistency was evaluated and in case of discrepancy solved by a third rater or a review and discussion of the data. A change in food choice was also considered a primary measure.

#### 2.4.5. Physical Activity and Sleep Pattern

Daily steps and minutes of sleep were also outcomes of interest. For research purposes, we considered a week as valid in the register when it had ≥4 complete days, defined as those with ≥10 h of any type of physical activity [49]. Fitbit non-wear time was defined as a period of ≥60 min with no steps registered [50]. 

### 2.5. Statistical Analysis

We performed a descriptive analysis for demographic variables comparing the groups at baseline (chi-square test or Fisher’s exact, Student’s *t* test or Mann–Whitney’s U test, see Table 1) to control for possible confounding variables (age, sex, IQ, handedness, economic income). Other relevant variables were also controlled (physical activity, initial motivation for treatment, hospital of provenance). 

For the evaluation of the effectiveness of treatment on anthropometrics, cognition, and rating scales, we performed mixed ANOVA with time as a 2-level within-subjects factor (T0, T1) and the group as an inter-subject condition. We used an intention-to-treat (ITT) approach and list-wise deletion was applied, analyzing the available data without imputing the unknown values. Multiple comparisons were tested by Bonferroni correction.

For longitudinal data (food choice, physical activity, and sleep patterns over 6 weeks), linear mixed models (LMM) with a random intercept and slope were used. Group and group per time interaction were set as fixed effects, and the subject’s ID was considered as a random effect in all these models.

Finally, following the Consort guidelines, [32] we ran an additional analysis on treatment adherence by comparing baseline characteristics between participants who reached a satisfactory training level (≥75%, completers) and who did not (<75%, non-completers).

All statistical tests were carried out using a two-sided test with the significance level set at 5% with the software Statistical Package for the Social Sciences (SPSS, version 24.0).

## 3. Results

Descriptive data and group comparison at baseline are shown in Table 1. No significant differences in baseline characteristics were found between groups, which were equally balanced for age, sex, handedness, IQ, economic status, hospital, BMI, physical activity, and motivation for treatment. Groups did not differ in any anthropometric and diet measures, neuropsychological variables, or rating scales (except for internalizing symptoms on CBCL, *p* = 0.03) (see Appendix A). Intention-to-treat analysis showed similar results but without any differences for internalizing symptoms on CBCL (see Appendix A). 

### 3.1. Effects of EF Training on Primary Outcomes 

#### 3.1.1. Cognition, Anthropometric Measures, and Food Choice

Changes from T0 to T1 in cognition, BMI, waist circumference (WC), and Kidmed questionnaire did not significantly differ between the two conditions, as shown in Table 2a. However, some within-group changes were found. There was a significant decrease over time in BMI in both groups (F_(1,34)_ = 8.81; *p* = 0.005; η_p_^2^ = 0.206) and also a reduction in waist circumference (F_(1,33)_ = 8.22; *p* = 0.007; η_p_^2^ = 0.199). ITT analysis showed similar results (see Appendix A).

#### 3.1.2. Evolution of Food Choice across Six-Week Training

Descriptive data for food choice are shown in Appendix A. Results from mixed effects models (Table 2b) suggested that there were no significant intervention group effects on food choice (β = 0.387 (CI −1.079 to 1.853), SE = 0.715, *p* = 0.593). However, there was a significant effect over time showing a tendency towards worse food choices in both groups (β = −0.926 (CI −1.806 to −0.045), SE = 0.427, *p* = 0.040). ITT analyses revealed similar results, although the effect of time on food choices was not significant (see Appendix A).

### 3.2. Effects of EF Training on Secondary Outcomes: Post Hoc Analyses

#### 3.2.1. Cognitive Subdomains

As shown in Table 3, significant improvements over time were found in attention and speed, working memory, cognitive flexibility, inhibitory control, and planning in both groups. Concrete benefits in attention and speed were detected in WNV spatial span forward (F_(1,34)_ = 6.76, *p* = 0.014, η_p_^2^ = 0.166), FDT reading time (F_(1,34)_ = 6.76, *p* = 0.014; η_p_^2^ = 0.166), counting time (F_(1,34)_ = 17.85, *p* = 0.000; η_p_^2^ = 0.344), and CPT reaction time (F_(1,34)_ = 12.03, *p* = 0.001; η_p_^2^ = 0.261). Benefits in working memory were found in WNV spatial span backward (F_(1,34)_ = 6.73, *p* = 0.014; η_p_^2^ = 0.166) and in spatial span forward (F_(1,34)_ = 5.06, *p* = 0.031; η_p_^2^ = 0.129), whereas improvements in cognitive flexibility were shown in CCTT-II (F_(1,34)_ = 10.23, *p* = 0.003; η_p_^2^ = 0.231) and FDT shifting time (F_(1,34)_ = 61.02, *p* = 0.000; η_p_^2^ = 0.642). Gains in inhibitory control appeared in FDT choosing time (F_(1,34)_ = 39.74, *p* = 0.000; η_p_^2^ = 0.539), go–no go correct responses (F_(1,32)_ = 6.65, *p* = 0.015, η_p_^2^ = 0.172) and commissions (F_(1,32)_ = 15.56, *p* = 0.000, η_p_^2^ = 0.327) and benefits in planning were detected in ToL total time (F_(1,34)_ = 7.98, *p* = 0.008; η_p_^2^ = 0.190).

Furthermore, there was a significant group by time interaction for WNV spatial span backward, in which the experimental group showed better performance (F_(1,34)_ = 5.25, *p* = 0.028, η_p_^2^ = 0.134; F_(1,34)_ = 6.73, *p* = 0.014, η_p_^2^ = 0.165); and for N-back-2, with better results in the control group (F_(1,34)_ = 4.55, *p* = 0.041, η_p_^2^ = 0.118; F_(1,34)_ = 5.06, *p* = 0.031, η_p_^2^ = 0.129).

After Bonferroni correction, significant improvements over time remained in attention and speed (FDT counting time, *p* = 0.000; and CPT reaction time, *p* = 0.001), cognitive flexibility (CCTT-II, *p* = 0.003; FDT shifting time, *p* = 0.000), and inhibitory control (FDT choosing time, *p* = 0.000; go–no go commissions, *p* = 0.000).

ITT analyses (see Appendix A) revealed similar results with some exceptions: there was no significant group-by-time interaction in WNV spatial span backward and N-back-2. The improvement in FDT reading over time was also non-significant. After Bonferroni correction, improvements over time remained significant in attention and speed (FDT counting time, *p* = 0.000), cognitive flexibility (CCTT-II, *p* = 0.000; FDT shifting time, *p* = 0.000), inhibitory control (FDT choosing time, *p* = 0.000; go–no go commissions, *p* = 0.000), and planning (ToL time, *p* = 0.001).

#### 3.2.2. Rating Scales

As displayed in Table 4a, changes from T0 to T1 did not significantly differ between the two conditions in rating scales, as there were no significant effects of time by group interaction on any measures. However, some within-group changes were found in both groups. There was a significant improvement over time in BRIEF-2 subscales: cognitive (F_(1,34)_ = 9.94, *p* = 0.003, η_p_^2^ = 0.226), emotional (F_(1,34)_ = 5.26, *p* = 0.028, η_p_^2^ = 0.134), and behavioral subscales (F_(1,34)_ = 5.46, *p* = 0.026, η_p_^2^ = 0.138), and also better scores in children-reported quality of life in PedsQl (F_(1,34)_ = 4.80, *p* = 0.035, η_p_^2^ = 0.124), increased self-esteem (F_(1,31)_ = 10.80, *p* = 0.003, η_p_^2^ = 0.258) and social support on SPPC (F_(1,30)_ = 4.18, *p* = 0.05, η_p_^2^ = 0.122), and decreased internalizing (F_(1,34)_ = 5.65, *p* = 0.023, η_p_^2^ = 0.142) and externalizing symptoms on CBCL (F_(1,34)_ = 4.74, *p* = 0.037, η_p_^2^ = 0.122). After Bonferroni correction, significant improvements over time remained in the BRIEF-2 cognitive subscale (*p* = 0.003) and self-esteem on SPPC (*p* = 0.003). 

ITT analysis showed similar results but the improvement in time in BRIEF-2 behavioral subscale and CBCL externalizing symptoms did not reach significance (see Appendix A). Moreover, none of the results remained significant after the Bonferroni correction. 

#### 3.2.3. Evolution of Healthy Habits (Physical Activity and Sleep Patterns)

Descriptive data for physical activity and sleep patterns over the 6 weeks are shown in Appendix A. Results from mixed effects models (Table 4b) suggested that there were no significant intervention group effects on physical activity (β = 9.042 (CI −1723.231 to 1741.315), SE = 847.632, *p* = 0.992) and sleep patterns (β = −0.664 (CI −23.242 to 21.915), SE = 11.095, *p* = 0.953). However, there was a significant effect over time, showing a reduction of sleep time for both groups (β = −18.151 (CI −33.242 to −3.061), SE = 7.416, *p* = 0.020). ITT analyses revealed similar results (see Appendix A).

### 3.3. Additional Analysis of Treatment Adherence

As shown in Table 5, of the 46 participants initially randomized (experimental group, n = 27; control group, n = 19), 36 completed satisfactory training (≥75% of sessions completed), which means a global adherence of 78.26%. Considering the group allocation, the experimental group showed an adherence of 70.3% and the control group of 89.17%. These differences did not reach statistical significance (*p* = 0.160). Independently of the allocation group, baseline characteristics of completer- and non-completer participants differed in several aspects. Completer-participants showed higher physical activity levels (*p* = 0.004), higher estimated visual IQ (*p* = 0.006), and increased motivation for treatment (*p* = 0.048). Furthermore, a trend for significance was found in the child-reported quality of life (PedsQl, *p* = 0.076) and perceived social support (SPPC, *p* = 0.083), with higher scores in completer participants.

## 4. Discussion

This study is one of the first to evaluate the impact of executive function training in children with obesity, the main objective being to present its effects on BMI, food choice, and cognition. The results revealed no significant differences between the experimental group and the active control group on these measures, but some within-group changes were found in BMI and waist circumference, which decreased over time in both. Similarly, post hoc analyses of specific cognitive subdomains revealed significant changes over time in attention and speed, working memory, cognitive flexibility, inhibitory control, and planning. These improvements did not significantly differ between groups, except for visual working memory, in which the experimental group performed better. After a restrictive correction, the effects remained still significant in several tests of attention and speed, cognitive flexibility, and inhibitory control. Regarding food choice, while results obtained from the Kidmed questionnaire indicated no changes post-training, the analysis of the daily food register showed a tendency for worse food choices for both groups over time.

All these findings imply that the executive function training with increasing difficulty did not produce stronger BMI decreases or better food decisions than the executive function training with the same difficulty, contrary to what we hypothesized. A plausible explanation is that we overestimated the difference in the executive function requirements between groups and they conducted a similar intervention, with other shared components that could also account for the observed effect. Therefore, we cannot conclude that the executive function training was directly linked with the observed effects, because of this limitation of the study [51]. Furthermore, another plausible explanation, according to previous research examining this issue, is that the most consistent effects on weight loss are seen after food-specific inhibition training, in contrast to other types of cognitive training [52]. In this regard, although we include inhibitory control in our training, it was designed with non-food-specific stimuli. Similarly to ours, a previous study with similar training characteristics (working memory and inhibitory control training with non-specific stimuli over 6 weeks) found no changes in BMI after training [22]. Regarding food decisions, some studies have revealed the potential of cognitive training to modify eating behavior (i.e., food choices in the laboratory) [52] but, typically, these studies measure food choice in a laboratory after relatively short interventions. In this sense and to date, there are no studies of cognitive training with a measure of food choices such as ours.

Regardless of the group, there was a reduction in BMI and waist circumference after training and a worsening in food choice over the training period. This was surprising, as it was expected that a decrease in weight measures should be associated with better food choices, especially if physical activity levels remained stable. We have some plausible explanations that are not mutually exclusive. First, participants could have started with high levels of intrinsic motivation because of the novelty of intervention, [53] showing better food choices at the beginning and then returning to their previous diet because of habituation to the stimuli. Second, initially, being monitored by an external health professional could have produced some discomfort in consuming certain types of food or quantities; therefore, they could have modified their consumption in a more or less conscious way. Third, pre-intervention physical activity levels may have been lower, and then they increased because of wearing an activity tracker and being monitored. This may explain why participants reduced their BMI and waist circumference despite the worsening of food choices. 

Regarding the secondary aim of the study, which was to evaluate the potential effect of cognitive training on children’s real-life executive functioning, emotional state, and quality of life, some within-group improvements were found in executive functioning, children-reported quality of life, perceived self-esteem, social support, internalizing, and externalizing problems in both groups. Of note, some of these improvements (the cognitive subdomain of executive functioning and perceived self-esteem) remained significant after a restrictive statistical correction. Again, the executive function enhancement alone did not explain these changes, but the whole intervention had a positive effect on several real-life measures. These results are aligned with sundry reviews conducted with cognitive training studies. Simons et al. [21] described better performance in trained tasks, with fewer effects on closely related tasks (near-transfer) and even less on distantly related tasks (far-transfer). Other authors concluded that far-transfer effects on other cognitive functions and real-world measures have been limited or inconsistent throughout studies [20,21,33,54].

Overall, we found that the executive function enhancement alone did not explain these changes, as there were no significant differences between groups. It might be that the control task training could also produce some benefits, so we found that the whole intervention had a positive impact on BMI and WC, several executive functions, and some measures of emotional state and quality of life, although being from the executive training group or being from the active control cognition did not explain differences. Some previous studies with active control groups using the Cogmed and Cognifit systems also found significant changes in several cognitive domains in both groups, suggesting that the low-intensity practice could also produce some benefits [34,55,56]. Preiss et al. [56] even described significant far-transfer improvements in both groups on the Cognitive Failures Questionnaire (CFQ) [57] and a trend for significance on the Dysexecutive Questionnaire (DEX) [58], both being measures of cognitive functioning in real life and, in this sense, comparable to our results in BRIEF for children. Related to the results of weight measures, it has been well documented that childhood obesity tends to worsen over time, as BMI in these children increases more exponentially than that in their normal-weight peers, according to the growth curves of the Clinical Growth Charts of the Centers for Disease Control and Prevention [59]. Therefore, our intervention had a positive short-term impact by altering the typical course of this health condition.

As said before, there were positive effects on several measures of emotional state, quality of life, and self-esteem after the intervention. These results are probably the most interesting finding, as there are no similar studies of cognitive training assessing these factors in children with obesity. However, there is some previous literature with non-clinical school samples. De Voogd et al. [60] found no differences in anxiety, depression, self-esteem, or socio-emotional behavioral problems after visual attention training. Hitchcoch and Westwell [61] showed similar results with no differences in socio-emotional behavioral difficulties, and Roberts et al. [62] did not find significant results either after applying Cogmed. All these studies applied exclusively cognitive training; thus, it seems reasonable that cognitive training may not be entirely responsible for the observed changes in our study. Although it was not specifically tested because it was not the aim of the study, it could be possible that other intervention components (i.e., psychoeducation, monitoring food register, wearing an activity tracker) account for the effects. In fact, a very recent study in children with obesity with multiple interventions (health education, physical exercise, and diet control) observed an improvement in psychological QoL in the intervention group, with higher psychological, pubertal, and total QoL all being higher in this group [63].

Finally, regarding treatment adherence, the global rate was 78.26% (70.3% in the experimental group and 89.17% in the control group). These differences did not reach statistical significance, although there was a trend for worse compliance in the experimental group. These findings are quite different from the average attrition rate in Cogmed trials, as typically studies lost 10% of the participants in the training group and 11% of the participants in the control group [64]. However, it has been documented that in pediatric obesity management, attrition could be as high as 80%, and 30 to 40% is common [65]. Moreover, typical Cogmed trials do not include other cognitive training programs, which require even more effort and commitment for participants. In this sense, our data are in line with those of other studies in this specific population. 

Additionally, in our sample, some baseline characteristics differed between completers and non-completer participants regardless of the group. The first showed higher physical activity levels, higher estimated visual IQ, and higher motivation for treatment. Furthermore, a trend was found for higher scores in self-reported quality of life and perceived social support. Previous literature reviews on factors related to attrition in pediatric obesity have revealed some important insights. For instance, Dhaliwal et al. [66] revealed that attrition was higher in older children (≥12 years old), whereas sex and baseline weight status did not predict it. Other factors explored among studies but with inconsistent results were internalizing and externalizing behavior, lower self-esteem, social problems, school problems, motivation, socioeconomic status, expectations of treatment, caregiver-rated quality of life, health status, ethnicity, home environment, success in BMI reduction during treatment, or lifestyle factors. Of note, heterogeneity is very high, and there were no clear answers on which factors contributed to dropout and what changes might improve retention [67]. Our results contribute to expanding the literature on which individual factors could be related to treatment adherence and convey the importance of examining this issue in future studies. They also pointed to the importance of perceived social support for treatment adherence.

### 4.1. Strengths and Limitations

This study has some strengths and some limitations. The main strengths include the three-fold intervention (executive function training, psychoeducation, and feedback on activity/sleep tracking) and its attractive delivery to children (it was implemented through an iPad, receiving feedback every day). Additionally, we evaluated transfer effect, which is an issue in most classical studies applying cognitive training that had focused primarily on assessing neuropsychological outcomes [68]. Moreover, we had a very good measure of food choice (photos of food intake), which overcomes traditional limitations of self-reported questionnaires such as bias problems [69,70]. Finally, we conducted the study with an active control group, which constitutes a more rigorous design, and makes the groups more comparable, potentially equating confounding factors and reducing motivational differences [21]. Nevertheless, active control training was selected ad hoc for our study, and the difference from experimental training was not tested before.

Regarding other limitations of the study, it was somewhat limited in statistical power due to the sample-size determination, which was based on previous literature in the field (for more details see Sanchez-Castañeda et al. [30]). In this sense, we could not introduce more explanatory factors in the statistical models to account for the observed variability over time, as this may have introduced bias in the effect size estimation. For this reason, larger randomized controlled trials with both active and passive control groups are worth considering. 

### 4.2. Conclusions and Future Research

Obesity treatment remains a public health challenge. Our results contribute to expanding the literature on which elements may be useful to add to multicomponent interventions for weight loss. Future studies with greater statistical power are still needed to better determine which of these elements account for the most variability and which individual factors predict a better response to treatment, as well as the type of stimuli that should be used in these interventions to improve their efficacy. In addition, it is still necessary to have more ecological measures of food intake that better reflect the daily life of the patient. In this sense, studies should continue examining how to assess diet more validly and reliably. Finally, more long-term intervention studies in younger patients with obesity should be promoted for preventing chronicity in adulthood, and to prevent developing any pathology. 

## Figures and Tables

**Figure 1 brainsci-13-00346-f001:**
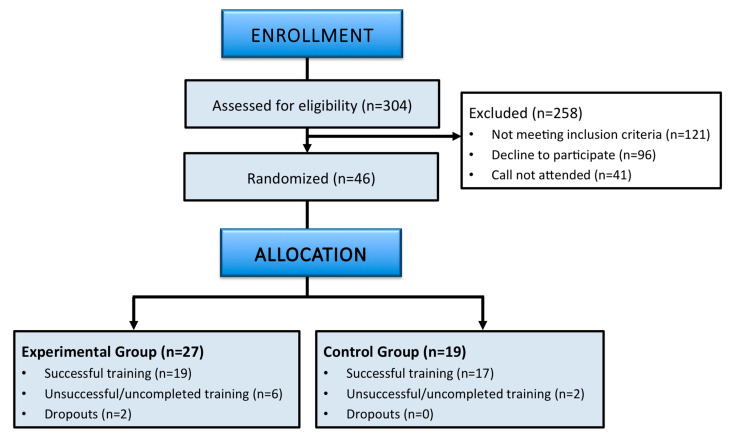
Flow diagram of the study recruitment with the sample according to CONSORT reporting guidelines (Schulz et al. 2010).

**Table 1 brainsci-13-00346-t001:** Baseline characteristics and group comparison.

	Experimental Group			Control Group			Group Comparison	
	N	Mean	SD	N	Mean	SD	*T/U*	Sig.
Age (years)	19	10.32	1.108	17	10.65	0.93	130.500	0.302
BMI (percentile)	19	98.63	0.83	17	98.52	0.76	−0.437	0.665
Physical activity (hours)	19	2.72	1.81	17	3.21	4.38	147.500	0.652
Visual IQ (scalar score)	19	10.68	2.31	17	10.35	2.15	135.500	0.403
Verbal IQ (scalar score)	19	11.42	2.80	17	10.76	2.25	−0.770	0.447
Motivation for treatment	19	6.31	0.749	16	6.63	0.719	114.000	0.217
	N			N			X^2^	Sig.
Sex (F:M)	5:14			8:9			1.673	0.196
Handedness (R:L)	18:1			15:2			-	0.593 ^†^
Economic income (I:II:III:IV:V)	6:7:1:2:1			3:3:4:4:0			5.485	0.219 ^††^
Hospital (CST: SJD)	14:5			12:5			.043	0.836

^†^ Fisher exact test. ^††^ Fisher–Freeman–Halton Test.

**Table 2 brainsci-13-00346-t002:** EF training effects on primary outcomes: (a) cognition and anthropometric pre-post measures and (b) evolution of food choice across the 6-week training period.

		Main Effects									Group-by-Time Interaction	
(a) Pre-Post Primary Outcomes		n	Pre-Test		Post-Test	Group			Time			
		Mean	(SD)		Mean (SD)	F (df); *p*		ηp^2^	F (df); *p*	ηp^2^	F (df); *p*	ηp^2^
Cognition (z score)	EG	19	0.054 (0.430)		0.107 (0.457)	*F*(_1,34_) = 1.50; 0.229		0.042	*F*(_1,34_) = 0.004; 0.949	0.000	*F*(_1,34_) = 1.22; 0.277	0.035
	CG	17	−0.061 (0.444)		−0.121 (0.453)							
BMI	EG	19	29.29 (3.18)		28.75 (2.99)	*F*(_1,34_) = 0.19; 0.662		0.006	*F*(_1,34_) = 8.81; **0.005**	0.206	*F*(_1,34_) = 0.72; 0.402	0.021
	CG	17	29.70 (4.22)		29.40 (4.11)							
WC	EG	19	91.94 (9.18)		89.60 (7.87)	*F*(_1,33_) = 0.22; 0.640		0.007	*F*(_1,33_) = 8.22; **0.007**	0.199	*F*(_1,33_) = 2.60; 0.116	0.073
	CG	16	92.53 (9.57)		91.44 (9.66)							
Kidmed	EG	16	7.11 (1.88)		6.13 (1.86)	*F*(_1,29_) = 0.00; 0.964		0.000	*F*(_1,29_) = 1.19; 0.284	0.040	*F*(_1,29_) = 2.21; 0.148	0.071
	CG	15	6.71 (2.02)		6.60 (2.19)							
**(b) Six-Week Training Primary Outcomes** ^†^			**Fixed Effects**					**Random Effects**				**Model Fit**
			**Est/Beta**	**SE**		**t**	** *p* **	**Param.**	**Covariance**	**SE**	**Sig.**	**AIC/BIC**
Kidmed Modified ^††^	Intercept		4.559	0.598	3.328 to 5.792	7.620	0.000	Residual	1.804	0.258	0.000	592.548/
	Group		−0.899	0.989	−2.923 to 1.131	−0.908	0.372	Intercept + Time				604.590
	Time		−0.926	0.427	−1.806 to −0.045	−2.166	**0.040**	[subject] UN (1,1)	4.690	1.831	0.010	
	Group x time		0.387	0.715	−1.079 to 1.853	0.541	0.593	UN (2,1)	−2.335	1.228	0.057	
								UN (2,2)	1.391	0.954	0.145	

Note. ^†^ Mixed-effects model parameters for food choice. ^††^ Scores could go from 0 to 15, larger scores mean better food choices. *F*, mixed ANOVA. Bold values indicate statistical significance (*p* < 0.05). Abbreviations: BMI = body mass index; CG = control group; EG = experimental group; WC = waist circumference.

**Table 3 brainsci-13-00346-t003:** EF training effects on secondary outcomes: post hoc analyses of cognitive subdomains.

					Main Effects				Group by Time Interaction	
		n	Pre-Test	Post-Test	Group		Time			
			Mean (SD)	Mean (SD)	F (df); *p*	η_p_^2^	F (df); *p*	η_p_^2^	F (df); *p*	η_p_^2^
Cognitive Domains										
Attention and Speed										
WISC-V Digit Span	EG	19	5.63 (1.38)	5.63 (0.90)	*F*_(1,34)_ = 0.61; 0.439	0.018	*F*_(1,34)_ = 0.51; 0.480	0.015	*F*_(1,34)_ = 0.51; 0.480	0.015
Forward	CG	17	5.29 (0.77)	5.53 (0.62)						
WNV Spatial Span	EG	19	5.74 (1.15)	6.47 (1.02)	*F*_(1,34)_ = 0.87; 0.357	0.025	*F*_(1,34)_ = 6.76; **0.014**	0.166	*F*_(1,34)_ = 0.84; 0.366	0.024
Forward	CG	17	5.65 (1.22)	6.00 (1.00)						
CCTT Part I Time	EG	19	25.89 (11.16)	22.37 (8.93)	*F*_(1,34)_ = 0.08; 0.781	0.002	*F*_(1,34)_ = 4.12; 0.050	0.108	*F*_(1,34)_ = 0.29; 0.597	0.008
	CG	17	24.29 (9.98)	22.24 (10.45)						
FDT Reading Time	EG	19	25.79 (4.21)	24.42 (4.06)	*F*_(1,34)_ = 0.34; 0.562	0.010	*F*_(1,34)_ = 6.76; **0.014**	0.166	*F*_(1,34)_ = 0.54; 0.467	0.016
	CG	17	26.29 (4.40)	25.53 (4.58)						
FDT Counting Time	EG	19	33.05 (5.67)	30.68 (5.00)	*F*_(1,34)_ = 0.92; 0.343	0.026	*F*_(1,34)_ = 17.85; **0.000** *	0.344	*F*_(1,34)_ = 0.25; 0.622	0.007
	CG	17	35.06 (6.12)	32.06 (5.63)						
CPT Detectability	EG	19	55.00 (9.32)	53.79 (8.69)	*F*_(1,34)_ = 0.39; 0.535	0.011	*F*_(1,34)_ = 1.98; 0.168	0.055	*F*_(1,34)_ = 0.01; 0.915	0.000
	CG	17	56.94 (9.22)	55.53 (9.72)						
CPT Omissions	EG	19	49.32 (6.79)	51.00 (7.89)	*F*_(1,34)_ = 3.75; 0.061	0.099	*F*_(1,34)_ = 1.45; 0.231	0.042	*F*_(1,34)_ = 0.04; 0.853	0.001
	CG	17	54.47 (9.25)	56.76 (14.03)						
CPT Reaction Time	EG	19	47.42 (6.44)	49.68 (8.37)	*F*_(1,34)_ = 1.15; 0.291	0.033	*F*_(1,34)_ = 12.03; **0.001** *	0.261	*F*_(1,34)_ = 2.03; 0.164	0.056
	CG	17	48.76 (7.09)	54.18 (12.37)						
Working Memory										
WISC-V Digit Span	EG	19	4.42 (0.84)	4.74 (0.81)	*F*_(1,34)_ = 2.97; 0.094	0.080	*F*_(1,34)_ = 0.69; 0.412	0.020	*F*_(1,34)_ = 0.69; 0.412	0.020
Backward	CG	17	4.18 (1.02)	4.18 (0.95)						
WNV Spatial Span	EG	19	5.21 (0.71)	6.16 (1.07)	*F*_(1,34)_ = 1.72; 0.199	0.048	*F*_(1,34)_ = 6.73; **0.014**	0.165	*F*_(1,34)_ = 5.25; **0.028**	0.134
Backward	CG	17	5.29 (1.21)	5.35 (0.99)						
N-Back (1-back)	EG	19	5.08 (1.00)	4.84 (1.28)	*F*_(1,34)_ = 0.10; 0.758	0.003	*F*_(1,34)_ = 0.11; 0.741	0.003	*F*_(1,34)_ = 0.98; 0.328	0.028
	CG	17	5.00 (1.16)	5.12 (0.84)						
N-Back (2-back)	EG	19	4.26 (1.33)	4.29 (1.84)	*F*_(1,34)_ = 0.39; 0.538	0.011	*F*_(1,34)_ = 5.06; **0.031**	0.129	*F*_(1,34)_ = 4.55; **0.041**	0.118
	CG	17	4.06 (1.58)	5.03 (0.79)						
N-Back (3-back)	EG	19	3.08 (0.93)	3.13 (1.36)	*F*_(1,34)_ = 0.02; 0.894	0.001	*F*_(1,34)_ = 0.15; 0.703	0.004	*F*_(1,34)_ = 0.02; 0.884	0.001
	CG	17	3.09 (1.08)	3.21 (1.15)						
Cognitive Flexibility										
CCTT Part II Time	EG	19	49.16 (14.24)	41.68 (11.02)	*F*_(1,34)_ = 0.14; 0.715	0.004	*F*_(1,34)_ = 10.23; **0.003** *	0.231	*F*_(1,34)_ = 0.26; 0.612	0.008
	CG	17	49.35 (11.94)	43.94 (8.31)						
FDT Shifting Time	EG	19	62.89 (12.77)	54.63 (12.33)	*F*_(1,34)_ = 0.00; 0.955	0.000	*F*_(1,34)_ = 61.02; **0.000** *	0.642	*F*_(1,34)_ = 0.18; 0.667	0.006
	CG	17	63.59 (10.60)	54.35 (9.31)						
Inhibitory Control										
FDT Choosing Time	EG	19	56.42 (11.99)	49.79 (9.03)	*F*_(1,34)_ = 0.06; 0.809	0.002	*F*_(1,34)_ = 39.74; **0.000** *	0.539	*F*_(1,34)_ = 0.17; 0.685	0.005
	CG	17	56.76 (7.60)	50.94 (9.20)						
GNG Correct	EG	19	214.58 (20.88)	226.68 (16.95)	*F*_(1,32)_ = 1.25; 0.272	0.038	*F*_(1,32)_ = 6.65; **0.015**	0.172	*F*_(1,32)_ = 0.16; 0.686	0.005
Responses	CG	15	210.00 (30.74)	216.06 (19.09)						
GNG	EG	19	32.89 (19.42)	24.26 (13.89)	*F*_(1,32)_ = 0.051; 0.823	0.002	*F*_(1,32)_ = 15.56; **0.000** *	0.327	*F*_(1,32)_ = 0.02; 0.895	0.001
Commissions	CG	15	33.25 (13.62)	27.75 (15.84)						
Planning										
ToL Total Move	EG	19	38.26 (15.19)	31.11 (15.59)	*F*_(1,34)_ = 0.01; 0.947	0.000	*F*_(1,34)_ = 2.14; 0.152	0.059	*F*_(1,34)_ = 0.17; 0.681	0.005
	CG	17	36.41 (24.21)	32.41 (7.47)						
ToL Total Time	EG	19	304.16 (117.94)	227.21 (106.61)	*F*_(1,34)_ = 0.01; 0.912	0.000	*F*_(1,34)_ = 7.98; **0.008**	0.190	*F*_(1,34)_ = 0.06; 0.806	0.002
	CG	18	302.18 (207.86)	237.65 (86.45)						

Note. *F*, mixed ANOVA. Bold values indicate statistical significance (*p* < 0.05) and * indicates Bonferroni-adjusted statistical significance (*p* < 0.003). Abbreviations: CCTT = Children’s Color Trail Test; CG = control group; CPT3 = Conners’ Continuous Performance Test 3; EG = experimental group; FDT = Five Digits Test; GNG: Go–No Go; ToL = Tower of London; WISC-V = Wechsler Intelligence Scale for Children (5th ed.); WNV = Weschler non-verbal.

**Table 4 brainsci-13-00346-t004:** EF training effects on secondary outcomes: (a) post hoc analyses of rating scales and (b) evolution of healthy habits across the 6-week training period.

					Main Effects Group Time						Group-by-Time Interaction	
(a) Rating Scales (Raw Score)		n	Pre-Test	Post-Test	F (df); *p*		η_p_^2^	F (df); *p*		η_p_^2^	F (df); *p*	η_p_^2^
BRIEF-2		19	52.74 (13.59)	49.79 (14.37)	*F*_(1,34)_ = 0.05; 0.820		0.002	*F*_(1,34)_ = 9.94; **0.003** *		0.226	*F*_(1,34)_ = 0.53; 0.474	0.015
Cognitive		17	52.65 (13.65)	47.94 (10.58)								
		19	25.68 (5.86)	24.11 (5.20)	*F*_(1,34)_ = 0.04; 0.840		0.001	*F*_(1,34)_ = 5.26; **0.028**		0.134	*F*_(1,34)_ = 0.31; 0.582	0.009
Emotional		17	26.53 (6.29)	23.94 (5.49)								
		19	19.16 (5.98)	17.32 (5.46)	*F*_(1,34)_ = 0.01; 0.907		0.000	*F*_(1,34)_ = 5.46; **0.026**		0.138	*F*_(1,34)_ = 0.17; 0.686	0.005
Behavioral		17	18.71 (3.97)	17.41 (3.95)								
PedsQl		19	76.72 (11.94)	81.64 (11.19)	*F*_(1,34)_ = 0.00; 0.953		0.000	*F*_(1,34)_ = 4.80; **0.035**		0.124	*F*_(1,34)_ = 0.33; 0.570	0.010
Child’s Total Score		17	77.94 (12.78)	80.81 (10.49)								
		19	73.92 (11.46)	79.29 (13.18)	*F*_(1,34)_ = 0.56; 0.461		0.016	*F*_(1,32)_ = 2.05; 0.162		0.057	*F*_(1,34)_ = 0.86; 0.361	0.025
Parents’ Total Score		17	72.72 (16.35)	73.87 (18.48)								
SPPC		18	20.50 (3.70)	21.74 (2.40)	*F*_(1,30)_ = 0.72; 0.402		0.023	*F*_(1,31)_ = 10.80; **0.003** *		0.258	*F*_(1,31)_ = 1.15; 0.292	0.036
Self-Esteem		15	18.93 (4.98)	21.82 (2.38)								
		18	21.25 (2.71)	21.64 (2.92)	*F*_(1,30)_ = 0.17; 0.680		0.006	*F*_(1,30)_ = 4.18; 0.050		0.122	*F*_(1,30)_ = 0.00; 0.948	0.000
Social Support		14	20.91 (2.70)	21.16 (1.98)								
CBCL		19	7.53 (4.11)	6.00 (4.64)	*F*_(1,34)_ = 4.09; 0.051		0.107	*F*_(1,34)_ = 5.65; **0.023**		0.142	*F*_(1,34)_ = 0.08; 0.778	0.002
Internalizing Symptoms		17	11.30 (5.73)	9.35 (7.89)								
		19	7.53 (6.25)	6.31 (6.40)	*F*_(1,34)_ = 0.11; 0.748		0.003	*F*_(1,34)_ = 4.74; **0.037**		0.122	*F*_(1,34)_ = 0.02; 0.904	0.000
Externalizing Symptoms		17	8.18 (4.45)	6.82 (4.98)								
**(b) Six-Week Training Secondary Outcomes** ^†^		**Fixed Effects**					**Random Effects**					**Model Fit**
		**Est/Beta**	**SE**	**95% CI**	**t**	** *p* **	**Param.**		**Covariance**	**SE**	**Sig.**	**AIC/BIC**
Daily Steps Sleep Time (min.)	Intercept	13,414.78	800.83	11,777.28 to 15,052.27	16.75	0.000	Residual Intercept + Time (subject) UN (1,1) UN (2,1) UN (2,2) Residual Intercept + Time (subject) UN (1,1) UN (2,1) UN (2,2)		4,433,933.35	581,326.30	0.000	3378.48/
	Group	−1357.17	1207.205	−3821.22 to 1106.88	0.112	0.270						3391.25
	Time	−230.94	50.32	−1357.39 to 895.50	−4.20	0.678			7,852,777.77	3,225,415.08	0.015	
	Group x time	9.04	847.63	−1723.23 to 1741.32	0.011	0.992			1,263,820.19	1,682,233.75	0.452	
									1,193,283.55	1,579,116.74	0.450	
	Intercept	494.91	8.73	477.122 to 512.69	56.66	0.000			792.493	105.84	0.000	1700.26/
	Group	−1.68	13.18	−28.525 to 25.16	−0.13	0.899						1712.83
	Time	−18.151	7.416	−33.242 to -3.06	−2.45	**0.020**			656.081	369.28	0.076	
	Group x time	−0.67	11.09	−23.242 to 21.92	−0.06	0.95			−165.994	270.63	0.540	
						3			167.384	259.45	0.519	

Note. ^†^ Mixed-effects model parameters for physical activity and sleep patterns. *F*, mixed ANOVA. Bold values indicate statistical significance (*p* < 0.05) and * indicates Bonferroni-adjusted statistical significance (*p* < 0.005). Abbreviations: BRIEF-2 = Behavior Rating Inventory of Executive Function 2; CBCL = Child Behavior Checklist; PedsQl = Pediatric Quality of Life Inventory; SPPC = Self-Perception and Social Support Profile for Children.

**Table 5 brainsci-13-00346-t005:** Comparison between completers (≥75%) and non-completers.

	Completers		Non-Completers		Group Comparison	
Accomplishment	N	%	N	%		
EG	19	70.37%	8	29.63%		0.160 ^†^
CG	17	89.17%	2	10.53%		
Individual Characteristics at Baseline						
	N	Mean (SD)	N	(SD)	X/T/U	*p*
Sex (M: F)	23:13	-	4:6	-	1.842	0.175
Age (years)	36	10.47 (1.03)	10	11.00 (0.816)	125.500	0.128
BMI (percentile)	36	98.58 (0.789)	10	98.32 (0.853)	0.89	0.380
WC (cm)	35	92.21 (9.23)	10	92.55 (14.01)	−0.09	0.928
Kidmed (raw score)	35	6.91 (1.93)	10	6.50 (1.96)	146.000	0.420
Physical activity (hours)	36	2.95 (3.24)	8	0.38 (1.06)	52.000	**0.004**
Daily screen time(hours)	36	2.14 (1.15)	8	2.39 (1.31)	120.500	0.469
Visual IQ (scalar score)	36	10.53 (2.21)	10	8.30 (1.83)	2.92	**0.006**
Verbal IQ (scalar score)	36	11.11 (2.54)	10	10.30 (2.87)	0.868	0.389
Motivation for treatment	35	6.48 (0.741)	8	5.75 (1.035)	82.500	**0.048**
Economic income	31	2.35 (1.19)	8	3.25 (1.58)	81.000	0.124
WISC-V						
Digit span forward	36	5.47 (1.13)	10	5.40 (0.84)	171.500	0.799
Digit span backward	36	4.31 (0.92)	10	4.10 (0.74)	160.500	0.581
WNV						
Spatial span forward	36	5.69 (1.17)	10	5.50 (1.08)	162.000	0.619
Spatial span backward	36	5.25 (0.97)	10	5.00 (1.70)	170.500	0.791
ToL						
Total move	36	37.39 (19.68)	10	33.60 (15.78)	179.000	0.979
Total time	36	303.22 (164.03)	10	328.20 (113.25)	130.500	0.187
CCTT						
Part I-seconds	36	25.14 (10.50)	10	30.80 (17.07)	145.500	0.358
Part II-seconds	36	49.25 (13.02)	10	55.20 (14.43)	137.000	0.252
**FDT**						
Reading	36	26.03 (4.24)	10	26.40 (4.86)	−0.238	0.813
Counting	36	34.00 (5.89)	10	36.50 (8.07)	−1.093	0.280
Choosing	36	56.58 (10.02)	10	57.70 (8.51)	162.500	0.641
Shifting		63.22 (11.64)	10	68.90 (15.92)	142.000	0.311
CPT (T-score)						
Detectability	36	55.92 (9.19)	10	57.50 (5.74)	176.500	0.926
Omissions	36	51.75 (8.34)	10	50.60 (6.52)	177.000	0.936
Hit reaction time	36	48.06 (6.69)	10	50.80 (9.09)	−1.059	0.205
N-back Correct responses						
1-back	36	5.01 (1.06)	10	4.60 (1.47)	148.500	0.388
2-back	36	4.17 (1.43)	10	3.70 (1.36)	143.500	0.327
3-back	36	3.08 (0.99)	10	3.10 (1.05)	−0.047	0.963
Go–No Go						
Correct responses	35	212.49 (25.56)	10	214.00 (20.07)	172.500	0.946
Commissions	35	33.06 (16.78)	10	34.10 (14.07)	167.000	0.827
BRIEF-2						
Cognitive	36	52.69 (13.42)	10	51.40 (13.22)	0.271	0.788
Emotional	36	26.08 (5.99)	10	25.20 (5.53)	0.419	0.677
Behavioral	36	18.94 (5.07)	10	17.70 (4.86)	154.000	0.487
PedsQl						
Child’s total score	36	77.29 (12.18)	10	68.95 (12.10)	113.500	0.076
Parents’ total score	36	73.36 (13.79)	10	70.21 (15.50)	0.622	0.537
SPPC						
Global self-esteem	33	19.79 (4.33)	10	17.60 (5.27)	117.000	0.164
Social support	33	21.10 (2.66)	10	19.09 (3.34)	101.500	0.083
CBCL						
Internalizing symptoms	36	9.30 (5.23)	10	10.80 (6.91)	163.000	0.650
Externalizing symptoms	36	7.83 (5.41)	10	6.70 (4.55)	0.605	0.548

Note. ^†^ Fisher exact test. Bold *p* values indicate statistical significance (*p* < 0.05).

## Data Availability

The data presented in this study are available on request from the corresponding author. The data are not publicly available due to privacy issues.

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
