# Peer review of "Influence of Executive Function Training on BMI, Food Choice, and Cognition in Children with Obesity: Results from the TOuCH Study"

_brainsci, 2023, doi:10.3390/brainsci13020346_

Round 1
Reviewer 1 Report
The manuscript is well written and results provide useful information for future studies. Limitations of the study are discussed and the results interpretation is good and clear.
I would suggest some minor revisions:
1. The experimental groups are identified in the abstract as “executive training” and “active control group”, while in the methods as “executive functions training” and “control task training”. Please be consistent, this could be confusing.
2. Please specify the ‘normal range’ for the QI (Lines 116-117).
3. Please provide information on the role of the family in supporting/guiding the children in the different interventions (e.g., how was the use of the iPad managed? Same for the psychoeducation and food register).
4. What does the authors mean when saying “daily pictures of the food intakes” (Line 155). Does this mean that they had to provide pictures of the meal?
5. Have the authors explored possible gender differences by comparing F and M groups?
Author Response
Answers to Reviewer 1:
- The experimental groups are identified in the abstract as “executive training” and “active control group”, while in the methods as “executive functions training” and “control task training”. Please be consistent, this could be confusing.
Thank you for your comment. We agree with the reviewer that this might drive to confusion. Therefore, we have reviewed the document and we have changed it consistently by using “executive functions training” and “control task training” through the whole document.
- Please specify the ‘normal range’ for the QI (Lines 116-117).
Again, we thank the reviewer for pointing that. We have now specified the normal range for QI, which is 80-120 (see line 118).
- Please provide information on the role of the family in supporting/guiding the children in the different interventions (e.g., how was the use of the iPad managed? Same for the psychoeducation and food register).
Participants provided pictures of all daily meals taken with the iPads and sent them directly through the mail installed on them. We were in contact though mobile phone with parents and children, especially if we saw that they did not send the required information or if we considered that the information was not reliable enough (for example, if they send only one meal per day). This mobile contact was done once or twice a week. Families could also reach our team by phone or mail if they experienced any technical issue, or if for any other reason (i.e. illness) they could not perform the daily training. We extended the methods section, including this information (see methods section, lines 150 to 153, and 162-163).
- What does the authors mean when saying “daily pictures of the food intakes” (Line 155). Does this mean that they had to provide pictures of the meal?
Indeed, participants provide pictures of all daily meals taken with the iPads and sent them directly through the mail installed on them. We understand that this was not clear enough in the manuscript. Therefore, we have made this more explicit in the text (see lines 160-161).
- Have the authors explored possible gender differences by comparing F and M groups?
We agree with the reviewer that this is an important issue. We actually included gender as covariate and this did not change substantially the results. Therefore, we did not include these additional analyses on the manuscript.
However, with the intention to answer to the reviewer’s concern, we have performed a General Lineal Model for repeated measures, with gender as the grouping variable and time as factor with two levels (pre vs. post-training measures) on primary outcomes, that is on BMI, waist circumference and cognition.
As Table 1 shows, there are significant differences over time (pre vs. post-training) in BMI and waist circumference; but the group, meaning the gender, does not behave significantly different. There is neither a significant interaction between group (gender) and time.
Table 1. Effect of gender on primary outcomes: cognition and anthropometric pre-post measures
|
|
|
Main effects |
Group by time interaction |
|||||
|
(a) Pre-post primary outcomes |
n Pre-test Mean (SD) |
Post-test Mean (SD) |
Group F (df); p |
Time F (df); p |
F (df); p |
|||
|
Cognition (z score) |
F M |
13 23 |
.054 (.430) -.061 (.444) |
.107 (.457) -.121 (.453) |
F= 1.06; .310 |
F= .000; .995 |
F= .001; .974 |
|
|
BMI
|
F M |
13 23 |
29.22 (4.58) 29.84 (3.02) |
28.64 (4.08) 29.46 (3.21) |
F= .179; .675 |
F= 9.66; .004 |
F= .588; .449 |
|
|
WC
|
F M |
13 23 |
90.12 (9.70) 93.44 (8.92) |
88.33 (9.14) 92.00 (8.42) |
F= 21.28; .265 |
F= 8.34; .007 |
F= .094; .761 |
|
Note. F, mixed ANOVA. Bold values indicate statistical significance (p < 0.05). Bonferroni corrected.
Abbreviations. BMI = body mass index; CG = control group; F = females; M = males, WC = waist circumference.

Reviewer 2 Report
Great idea. It would be good to repeat on a larger sample.
A praise worthy study. And, quite a few surprising results. Good basis for further research.
Here are several recommendations for corrections and improvement.
Although the article refers to the population of children who are already obese, the question of the role of primary prevention of obesity is unavoidable. Considering that possible interventions are widely discussed in the introduction, I recommend inserting sentence or two, regarding the possibilities of primary prevention, because it seems that this is the only real opportunity to change the current trend of increasing obesity globally. It is known that habits are formed in the first 1000 days of life, I would like to see a review of reference(s) related to this possibility of intervention; namely education of parents of infants and small children, education of pregnant women and future parents. Perhaps the window of opportunity opened much earlier. Given that previous as well as contemporary interventions are not successful, it is important to always emphasize this. For general culture and knowledge in this field of pediatric medicine.
Line 24:
Digitization of treatment supervision in view of new civilizational trends is clearly becoming our reality. On the other hand, we want to move children away from the monitor and reduce the time spent in the virtual world, it will be difficult to reconcile these two goals.
During monitoring, in addition to virtual contact via iPad, was there also traditional monitoring, contact with parents of another type such as a phone call or something else?
Line 38:
Is there more recent epidemiological data on the prevalence of childhood obesity than the one mentioned in 2016? Namely, the numbers have grown significantly since then.
Line 62:
It would be good to add some recent references that talk about this area, like for example:
- Jansen PW, Derks IPM, Mou Y, et al. Associations of parents' use of food as reward with children's eating behaviour and BMI in a population-based cohort. Pediatr Obes. 2020;15(11):e12662. doi:10.1111/ijpo.12662
- Tan CC, Lumeng JC. Associations Between Cool and Hot Executive Functions and Children's Eating Behavior. Curr Nutr Rep. 2018;7(2):21-28. doi:10.1007/s13668-018-0224-3
- Wang SD, Nicolo M, Yi L, Dunton GF, Mason TB. Interactions among Reward Sensitivity and Fast-Food Access on Healthy Eating Index Scores in Adolescents: A Cross-Sectional Study. Int J Environ Res Public Health. 2021;18(11):5744. Published 2021 May 27. doi:10.3390/ijerph18115744
Line 104:
I am interested in who gave written consent to participate in the study; only parents or parents AND children? It is usual for children from the age of 6 to be informed and give their consent.
Line 120-126:
In the flow diagram of the study recruitment, there is an unusually large number of children who did not meet the inclusion criteria (121/304). I'm interested in a little more detail about what those reasons were, given that today we know that most of these children have primary obesity, usually without comorbidities. It is also unfortunate that a large number (96/304) did not want to participate.
Line 167-168:
Anthropometric parameters include body mass, height, and waist circumference. Indexes were not mentioned; BMI nor Z-score. Has the Z-score been calculated? This is the standard in pediatric medicine, especially nutritional medicine.
Line 216 and 322-329:
It is a pity that there is no more detailed analysis on physical activity. It is at least as important, if not more so, than eating habits. It states 'any type of physical activity'; does this mean that ordinary walking is also acceptable/included, just estimating the number of steps, regardless of intensity? It is known that the degree of physical activity is important. It is not enough just to walk, rhythm and intensity are also important. A light walk cannot significantly contribute to the therapeutic effect, it does not represent an essential intervention. Maybe more for a psycho-emotional state, especially if it's in nature. I recommend to explain in more detail.
The results show the difference in absolute BMI values, which is mathematically significant. However, from a medical point of view, the dynamics of the Z-score for BMI is important, as already mentioned above. In this way, you can get a much better insight into the real degree of obesity and the importance of body mass dynamics.
Line 264:
The results about the tendency for worst food choice are really surprising! Benefit in performance, cognition, memory, speed, but not in the most important, decision-making.
However, later in the discussion, they were well elaborated, reasonable arguments, or hypotheses of possible reasons, were offered.
Line 282:
Correct typos. The sentence is unclear.
Line 297 and 299:
Again, surprising results; increase in inhibitory control but also increase in worse food choice. There is an important fact to note about the increase in self-esteem; it is to be hoped that this will have a long-term impact on changing lifestyles and eating habits.
Line 417:
Unfortunately, yes, we have to be satisfied even with the effect of stopping the rapid gain of body mass
Line 467:
Fix typo, iPad
Line 470-471;
Like before; the question is whether there was contact with parents and children apart from the virtual/digital activity and training companion? Photos of food intake are a very ecological measure, but I recommend emphasizing that the question of credibility still remains.
In general; is there a plan for long-term follow-up of these children due to the risk of developing an eating disorder, which is significant with any type of intervention in the child's menu? Of course, the biggest risk comes from going on a diet (usually fasting) without supervision. It often happens that these children continue with a disordered eating pattern and develop the full spectrum of eating disorder. If this is being considered and implemented, it is important to emphasize this, for the sake of educating the reader, raising awareness of the importance of supervision by the medical team, in the case of all types of nutritional interventions in children and adolescents. With supervision, the risk of developing an eating disorder is reduced to a minimum.
Author Response
Answers to Reviewer 2:
Although the article refers to the population of children who are already obese, the question of the role of primary prevention of obesity is unavoidable. Considering that possible interventions are widely discussed in the introduction, I recommend inserting sentence or two, regarding the possibilities of primary prevention, because it seems that this is the only real opportunity to change the current trend of increasing obesity globally. It is known that habits are formed in the first 1000 days of life, I would like to see a review of reference(s) related to this possibility of intervention; namely education of parents of infants and small children, education of pregnant women and future parents. Perhaps the window of opportunity opened much earlier. Given that previous as well as contemporary interventions are not successful, it is important to always emphasize this. For general culture and knowledge in this field of pediatric medicine.
We thank the reviewer for pointing that. We have now addressed this issue and expanded the literature according to that suggestion (see Discussion section, lines 502-505)
Line 24: Digitization of treatment supervision in view of new civilizational trends is clearly becoming our reality. On the other hand, we want to move children away from the monitor and reduce the time spent in the virtual world, it will be difficult to reconcile these two goals.
During monitoring, in addition to virtual contact via iPad, was there also traditional monitoring, contact with parents of another type such as a phone call or something else?
Thank you for this observation. It is true that, on the one hand, we want to reduce the screen time exposure; and, on the other hand, we want to take advantage of online intervention and of the adherence screens have for children. A possible solution would be, to involve parents more actively in the therapy (actually, they also had an active role in this study) in order to reduce the screen time for playing/entertainment and limit this exposure to therapeutic or educative purposes. So we have detailed this information on lines 150 to 153.
Regarding monitoring, we were in contact though mobile phone with parents and children, especially if we saw that they did not send the required information or if we considered that the information was not reliable enough (for example, if they send only one meal per day). This mobile contact was done once or twice a week. Families could also reach our team by phone or mail if they experienced any technical issue, or if for any other reason (i.e. illness) they could not perform the daily training. We extended the methods section, including this information (see methods section, lines 150 to 153, and 162-163).
Line 38: Is there more recent epidemiological data on the prevalence of childhood obesity than the one mentioned in 2016? Namely, the numbers have grown significantly since then.
Thanks for the comment. We are aware of this increased prevalence numbers, but we maintained data of 2016, as it was the last official data when we designed the research project. However, with the intention to answer to the reviewer’s concern, we have updated the data according to WHO at the Introduction and References sections (see lines 39-40, and 536-537).
Line 62: It would be good to add some recent references that talk about this area, like for example:
- Jansen PW, Derks IPM, Mou Y, et al. Associations of parents' use of food as reward with children's eating behaviour and BMI in a population-based cohort. Pediatr Obes. 2020;15(11):e12662. doi:10.1111/ijpo.12662
- Tan CC, Lumeng JC. Associations Between Cool and Hot Executive Functions and Children's Eating Behavior. Curr Nutr Rep. 2018;7(2):21-28. doi:10.1007/s13668-018-0224-3
- Wang SD, Nicolo M, Yi L, Dunton GF, Mason TB. Interactions among Reward Sensitivity and Fast-Food Access on Healthy Eating Index Scores in Adolescents: A Cross-Sectional Study. Int J Environ Res Public Health. 2021;18(11):5744. Published 2021 May 27. doi:10.3390/ijerph18115744
Again, we thank the reviewer for pointing that. We know that there are more updated studies, but we preferred to cite those that were relevant at the time we designed the study at the Introduction section. However, considering the suggestions of the reviewer, we have included the third reference in the text (see reference 15).
Line 104: I am interested in who gave written consent to participate in the study; only parents or parents AND children? It is usual for children from the age of 6 to be informed and give their consent.
The whole family was informed and gave their consent, children did it verbally. However, in legal terms, in Spain, parents’ are the ones that sign the written consent as legal tutors. We agree with the reviewer that it is important to make explicit on the manuscript that children also gave their verbal consent, we have therefore modified the text including this information (see lines 104-105).
Line 120-126: In the flow diagram of the study recruitment, there is an unusually large number of children who did not meet the inclusion criteria (121/304). I'm interested in a little more detail about what those reasons were, given that today we know that most of these children have primary obesity, usually without comorbidities. It is also unfortunate that a large number (96/304) did not want to participate.
We have just checked the reasons of exclusion and have found that 90% of children did not meet criteria for obesity but did it for overweight. Others (10%) could not participate for technical reasons (no Wi-Fi connection) or having diagnosis of a neurodevelopmental disorder, most commonly ADHD.
Line 167-168: Anthropometric parameters include body mass, height, and waist circumference. Indexes were not mentioned; BMI nor Z-score. Has the Z-score been calculated? This is the standard in pediatric medicine, especially nutritional medicine.
We agree with the reviewer, that the most standardized and widely used measures are BMI z-scores. However, we based our decision on specific literature, which supports the elimination of the BMIz as a measure of change in severe obesity (Dietz, 2017; https://doi.org/10.1542/peds.2017-2148), as it is a poor predictor of changes in total body fat (Vanderwall et al., 2018; https://doi.org/10.1186/s12887-018-1160-5).
However, with the purpose to answer the reviewer’s concern, we have performed a General Lineal Model for repeated measures, with training group (Experimental vs. Control) as the grouping variable and BMI z as a factor with two levels (pre vs. post-training).
As Table 2 shows, results are very similar by using BMI raw score or z score. There are significant differences over time (pre vs. post-training) in weight, concretely a weigh loss; but there are no significant differences between training groups, nor a time per group interaction.
Table 2. EF training effects on primary outcomes: comparison of BMI and BMIz measures
|
|
|
Main effects |
Group by time interaction |
|||||||
|
(a) Pre-post primary outcomes |
n Pre-test Mean (SD) |
Post-test Mean (SD) |
Group F (df); p |
Time F (df); p |
F (df); p |
|||||
|
BMI
|
EG CG |
19 17 |
29.29 (3.18) 29.70 (4.22) |
28.75 (2.99) 29.40 (4.11) |
F(1,34)= .19; .662 |
F(1,34)= 8.81; .005 |
F(1,34)= .72; .402 |
|||
|
BMIz
|
EG CG |
19 17 |
2.27 (0.249) 2.23 (0.254) |
2.21 (0.259) 2.21 (0.261) |
F(1,34)= .109; .743 |
F(1,34)= 21.20 .000 |
F(1,34)= .692; .411 |
|||
Note. F, mixed ANOVA. Bold values indicate statistical significance (p < 0.05). Bonferroni corrected.
Abbreviations. BMI = body mass index; CG = control group; EG = experimental group; WC = waist circumference.
Line 216 and 322-329: It is a pity that there is no more detailed analysis on physical activity. It is at least as important, if not more so, than eating habits. It states 'any type of physical activity'; does this mean that ordinary walking is also acceptable/included, just estimating the number of steps, regardless of intensity? It is known that the degree of physical activity is important. It is not enough just to walk, rhythm and intensity are also important. A light walk cannot significantly contribute to the therapeutic effect, it does not represent an essential intervention. Maybe more for a psycho-emotional state, especially if it's in nature. I recommend to explain in more detail.
The results show the difference in absolute BMI values, which is mathematically significant. However, from a medical point of view, the dynamics of the Z-score for BMI is important, as already mentioned above. In this way, you can get a much better insight into the real degree of obesity and the importance of body mass dynamics.
We agree with the reviewer that it is a pity that we could not run a more detailed analysis of physical activity. Fitbit system actually targets the intensity of the physical activity, grouping it into light (low), medium or intense (high). However, we found that data had not enough variability to detect significant differences between groups. Actually, most of the children’s physical activity was low-medium. Also, considering our sample-size, too many variables could drive to bias. However, if the reviewer considers if this information is relevant for the article, we could provide this information as supplementary material.
Regarding the Z score, as Table 2 shows, results are very similar by using BMI raw score or z score. There are significant differences over time (pre vs. post-training) in weight, concretely a weight loss; but there are no significant differences between training groups, nor a time per group interaction.
Line 264: The results about the tendency for worst food choice are really surprising! Benefit in performance, cognition, memory, speed, but not in the most important, decision-making.
However, later in the discussion, they were well-elaborated, reasonable arguments, or hypotheses of possible reasons, were offered.
We agree with the reviewer that those were surprising results and we appreciate that, from his/her point of view, we addressed on the manuscript reasonable arguments or hypotheses to this respect.
Line 297 and 299: Again, surprising results; increase in inhibitory control but also increase in worse food choice. There is an important fact to note about the increase in self-esteem; it is to be hoped that this will have a long-term impact on changing lifestyles and eating habits.
We agree with the reviewer that those were surprising results and appreciate that, from his/her point of view, we elaborated properly reasonable arguments.
Line 282: Correct typos. The sentence is unclear.
We thank the reviewer for pointing that. We have now corrected the mistake, changing n for ηp2 (lines 284-287).
Line 467: Fix typo, iPad
We thank again the reviewer for noticing this. We have corrected the text accordingly.
Line 470-471: Like before; the question is whether there was contact with parents and children apart from the virtual/digital activity and training companion? Photos of food intake are a very ecological measure, but I recommend emphasizing that the question of credibility still remains.
In general, is there a plan for long-term follow-up of these children due to the risk of developing an eating disorder, which is significant with any type of intervention in the child's menu? Of course, the biggest risk comes from going on a diet (usually fasting) without supervision. It often happens that these children continue with a disordered eating pattern and develop the full spectrum of eating disorder. If this is being considered and implemented, it is important to emphasize this, for the sake of educating the reader, raising awareness of the importance of supervision by the medical team, in the case of all types of nutritional interventions in children and adolescents. With supervision, the risk of developing an eating disorder is reduced to a minimum.
We agree with the reviewer that sending pictures could be biased if children do not send all the pictures of the intakes. For that reason, we only included in the study children whose parents were willing to actively participate and involve in the study as direct supervisors of the behavior of their children. This was also one of the main reasons for establishing the age-range, as children below 12 years old still depend and allow parents to participate on their tasks.
Regarding the follow-up, in order to prevent the development of any kind of eating disorders, we again agree with the reviewer that this is a very crucial point. Children participating in the study were also clinically followed by a specialist on children endocrinology in two third-level hospitals. This follow-up lasts until children reach adulthood, and would allow detecting any deviation from the norm and/or prevent or detect the development of any eating disorders. Therefore, following the reviewer’s suggestions, we have extended the discussion section (lines 510-512) in order to provide a deeper view of these possibilities and the different strategies we have developed to prevent or to detect them.
Finally, we have checked the whole manuscript carefully and also sent the document to our language reviewers to detect any language issue. We hope that this is clearer now. Thank you for your general suggestion.
